# Association between tumor mutation profile and clinical outcomes among Hispanic Latina women with triple-negative breast cancer

Alexander Philipovskiy[1]*, Alok K. Dwivedi[2], Roberto Gamez[3], Richard McCallum[4], Debabrata Mukherjee[5], Zeina Nahleh[6], Renato J. Aguilera[7], Sumit Gaur[1]

1 Division of Hematology-Oncology, Department of Internal Medicine, Texas Tech University Health Sciences Center El Paso, El Paso, Texas, United States of America, 2 Division of Biostatistics & Epidemiology, Department of Molecular and Translational Medicine, Paul L. Foster School of Medicine, Texas Tech University Health Sciences Center El Paso, El Paso, Texas, United States of America, 3 Department of Pathology, University Medical Center, Texas Tech University Health Sciences Center El Paso, El Paso, Texas, United States of America, 4 Division of Gastroenterology, Department of Internal Medicine, Texas Tech University Health Sciences Center El Paso, El Paso, Texas, United States of America, 5 Division of Cardiology, Department of Internal Medicine, Texas Tech University Health Sciences Center El Paso, El Paso, Texas, United States of America, 6 Department of Hematology-Oncology, Maroone Cancer Center, Cleveland Clinic, Florida, Weston, Florida, United States of America, 7 Border Biomedical Research Center, The University of Texas at El Paso, El Paso, Texas, United States of America

* alexander.philipovskiy@ttuhsc.edu

**Data Availability Statement:** All relevant data are within the paper.

## Abstract

Triple-negative breast cancer (TNBC) represents 15%–20% of all breast cancer types. It is more common among African American (AA) and Hispanic-Latina (HL) women. The biology of TNBC in HL women has been poorly characterized, but some data suggest that the molecular drivers of breast cancer might differ. There are no clinical tools to aid medical oncologists with decisions regarding appropriate individualized therapy, and no way to predict long-term outcomes. The aim of this study was to characterize individual patient gene mutation profiles and to identify the relationship with clinical outcomes. We collected formalin-fixed paraffin-embedded tumors (FFPE) from women with TNBC. We analyzed the gene mutation profiles of the collected tumors and compared the results with individual patient's clinical histories and outcomes. Of 25 patients with TNBC, 24 (96%) identified as HL. Twenty-one (84%) had stage III–IV disease. The most commonly mutated genes were *TP53*, *NOTCH1*, *NOTCH2*, *NOTCH3*, *AKT*, *MEP3K*, *PIK3CA*, and *EGFR*. Compared with other international cancer databases, our study demonstrated statistically significant higher frequencies of these genes among HL women. Additionally, a worse clinical course was observed among patients whose tumors had mutations in *NOTCH* genes and *PIK3CA*. This study is the first to identify the most common genetic alterations among HL women with TNBC. Our data strongly support the notion that molecular drivers of breast cancer could differ in HL women compared with other ethnic backgrounds. Therefore, a deeper understanding of the biological mechanisms behind *NOTCH* gene and *PIK3CA* mutations may lead to a new treatment approach.

**Funding:** RJA was supported by a Research Centers in Minority Institutions (RCMI) program grant (2U54MD007592) to the Border Biomedical Research Center (BBRC) at UTEP from the National Institute on Minority Health and Health Disparities, a component of the National Institutes of Health. Funding was also provided by Texas Tech University Medical Sciences Center Department of Internal Medicine Seed Founding program 2018–2019.

**Competing interests:** The authors have declared that no competing interests exist.

# Introduction

Triple-negative breast cancer (TNBC) is characterized by a lack of steroid hormone receptor expression, such as estrogen (ER) and progesterone (PR), and also by the absence or low expression of the tyrosine kinase receptor HER2. It represents approximately 15%–20% of all newly diagnosed breast cancer cases in the United States [1]. Typically, TNBC has an aggressive natural course characterized by the rapid development of chemotherapy resistance, higher recurrence rates, and poor outcomes. Because of the lack of targetable receptors, chemotherapy remains the mainstay of treatment for patients with TNBC.

In the past decades, significant progress had been achieved in understanding the biology of TNBC [2, 3]. TNBC is a heterogenic disease that can be additionally subdivided into at least four distinct subtypes based on tumor gene expression profiles. These subtypes are characterized by different clinical courses and resistance to chemotherapy [2–5]. The basal-like subtype 1 (BL-1) is usually characterized by a better progression-free survival (PFS) rate compared with the other subtypes. Pathological features of BL-1 tumors include high tumor grade and high Ki-67 proliferation index (>85%). Importantly, the BL-1 subtype is highly sensitive to chemotherapy, with a response rate approaching 60%. In contrast, the BL-2 subtype is clinically characterized by the worst PFS and early metastasis. BL-2 has the same pathological features as BL-1 but is resistant to conventional chemotherapy. Two other subtypes, mesenchymal subtype (M) and luminal androgen receptor subtype (LAR), are characterized by a relatively low Ki-67 index (<50%) and an indolent clinical course with very modest sensitivity to chemotherapy, and a response rate of 10%–20% [4]. Several studies have evaluated the role of numerous genetic alterations as prognostic markers for outcomes (*BRCA1*/*BRCA2* and *PIK3CA*/*AKT*/*MTOR*) and/or predictive markers for chemotherapy resistance (*TP53*/*PIK3CA*/*AKT*/*MTOR*, and *AR*) [6, 7]. Molecular profiling is a very promising tool to predict individual tumor response to chemotherapy. However, such an approach has not yet been validated in prospective clinical trials.

To date, there is no reliable tool to predict individual tumor response or resistance to chemotherapy and/or patients' outcomes other than direct response to neoadjuvant chemotherapy. Pathological complete response (pCR) to neoadjuvant chemotherapy is an accepted surrogate marker for favorable outcomes in patients with early-stage disease [8]. More recently, a tumor genetic profiling tool was described as a possible approach to predict the pCR to neoadjuvant chemotherapy (BA100) [9]. A robust predictive tool that can provide physicians with important information about tumor aggressiveness and enable individualization of treatment approaches would be highly desirable, especially for metastatic TNBC (mTNBC). It would be ideal, for instance, to identify which tumors are more chemoresistant and thus potentially require a multi-agent chemotherapy regimen while avoiding overtreatment in patients with less aggressive tumors. Furthermore, identifying unique targets that might be more prevalent in specific subtypes of TNBC or certain patient populations may help in developing novel and more personalized treatment approaches for mTNBC patients.

In this study, we sought to understand the prevalence of potentially targetable mutations in a group of HL women with TNBC. Notable differences in the incidence and mortality of breast cancer have been suggested among various racial and ethnic groups [10]. The age-adjusted incidence of breast cancer per 100,000 is around 128 for non-Hispanic white (NHW) women, 125 for African American (AA) women, and 92 for (HL) women [1, 11]. Importantly, multiple studies suggest that the prevalence of TNBC among HL women is slightly higher compared with NHW, approaching 23.1% [12–15]. Additionally, the onset of the disease occurs in women approximately 11 years younger than the average age reported for NHW and AA women [1, 11], while the overall breast cancer incidence among AA and HL populations has

continued to grow [1]. It is unclear whether there are underlying biological and genetic drivers of breast cancer that are more prevalent in HL women [16]. Specifically, there is a gap in our knowledge regarding the genetic mutation profiles of different racial/ethnic subgroups because few studies have addressed genetic diversity among HL women, especially those with mTNBC.

This study aimed to characterize the mutational profile of TNBC tumors in a HL population and its association with treatment response, and to identify whether there are recurrent mutations that could contribute to future therapeutic targeted studies.

## Materials and methods

### Patient population

The study protocol was reviewed and approved by the Texas Tech University Health Sciences Center El Paso (TTUHSC EP) Institutional Review Board before the commencement of the study. Due to the retrospective nature of the study, written informed consent was not required. All data/tissue samples were fully anonymized. In this study we retrospectively reviewed the clinical databases of the Texas Tech Breast Care Center and the University Medical Center in El Paso, Texas, from January 2012 to December 2019 to identify all patients with a diagnosis of stage II–IV TNBC who received treatment at our institution. Only newly registered cases were extracted from the databases. Any patients with incomplete data on outcomes such as overall survival (OS) or progression free survival (PFS) were excluded from the study.

### Pathologic assessment

Pathological diagnosis, hormonal status (ER and PR), and HER2 status were determined during the initial evaluation and before chemotherapy. Standard immunohistochemical (IHC) staining was used to determine hormonal receptor status. All tumors with less than 1% stained cells were considered to have a negative hormonal receptor status. HER2 status was evaluated by IHC staining only if it scored 0 or 1+. For specimens that scored 2+, fluorescence in situ hybridization was used for confirmation of HER2 negativity.

### Tumor genome sequencing

For this study, we retrospectively collected and analyzed the whole genome sequencing data (Foundation Medicine, FoundationoneCDX Cambridge, MA, USA) of 25 female patients with TNBC who were treated at the Texas Tech Breast Care Center from 2012 to 2019. Briefly, patients' DNA was extracted from FFPE samples. The assay employed a single DNA extraction method from routine FFPE biopsy or surgical resection specimens, 50–1000 ng of which underwent whole genome shotgun library construction and hybridization-based capture of all coding exons from 309 cancer-related genes, one promoter region, one non-coding (ncRNA), and select intronic regions from 34 commonly rearranged genes, 21 of which also include the coding exons. In total, the assay detected alterations in 324 genes. Using Illumina® HiSeq 4000 (Illumina, Inc. San Diego, CA, USA) platform-hybrid capture, selected libraries were sequenced to high uniform depth (targeting >500× median coverage with >99% of exons at >100× coverage). Sequence data were then processed using a customized analysis pipeline designed to detect all classes of genomic alterations, including base substitutions, indels, copy number alterations (amplifications and homozygous gene deletions), and selected genomic rearrangements (for details go to: https://assets.ctfassets.net/vhribv12lmne/4ZHUEfEiI8iOCk2Q6saGcU/b69f05b7fc06bf73e0aa1a6f2bee982b/F1CDx_TechInfo_10-09.pdf.

## Comparing our data with international databases

We compared the frequency of cancer gene mutations discovered in our study with previously published databases TCGA (The Cancer Genome Atlas), METABRIC (Molecular Taxonomy of Breast Cancer International Consortium), and COSMIC (Catalogue of Somatic Mutations in Cancer), as well as Chinese [17], and Thai studies [18] using z-tests for proportions. The data were downloaded from the cBioPortal for Cancer Genomics (https://www.cbioportal.org/study/summary?id=brca_metabric). TCGA cohort consisted of cancer genome data from primary breast cancer patients in the United States (https://portal.gdc.cancer.gov/projects?filters=%7B%22op%22%3A%22and%22%2C%22content%22%3A%5B%7B%22op%22%3A%22in%22%2C%22content%22%3A%7B%22field%22%3A%22projects.primary_site%22%2C%22value%22%3A%5B%22breast%22%5D%7D%7D%2C%7B%22op%22%3A%22in%22%2C%22content%22%3A%7B%22field%22%3A%22projects.program.name%22%2C%22value%22%3A%5B%22TCGA%22%5D%7D%7D%5D%7D). METABRIC cohort data were collected from primary breast cancer patients in the United Kingdom and Canada [19]. The COSMIC database is from the Cancer Genome Project at the Sanger Institute (https://cancer.sanger.ac.uk/cosmic/download). Cancer gene mutation data from TCGA database were selected only from breast invasive ductal carcinoma that was classified as PAM50 basal subtype, which is closely related to the triple-negative subtype of breast cancer, while data from METABRIC and COSMIC databases were selected from samples with negative ER, PR, and HER2, similar to this study.

## Mutation classification

Different mutations in *TP53* were classified to predict the effect on p53 protein function [20] by searching for missense mutations in the DNA-binding domain (DBD) and outside the DBD, as well as non-missense mutations (including splice, frameshift, and nonsense mutations).

## Outcome measures and statistical analysis

The primary clinical outcomes included PFS and OS. The PFS was defined from the date of treatment to the date of recurrence or last follow-up whereas the OS was defined from the date of treatment to the date of mortality or last follow up. Continuous variables were characterized using mean, standard deviation (SD), minimum, and maximum values, while categorical variables were summarized using frequency and percentages. The expression of each gene was categorized as present (1) or not present (0). The number of chemotherapy cycles was counted as well as the number of genes that were expressed per patient. The associations between gene groups and baseline characteristics were assessed using Fisher's exact test or unpaired t-test depending on the type of variable. The associations between gene groups and age at diagnosis and advanced stage were assessed using Fisher's exact test. A variable cluster analysis for categorical variables was performed to identify the clustering among the gene groups and related cluster scores. The optimum number of clusters was determined using the aggregation plot and mean adjusted Rand criteria, which indicated that the three clusters retained maximum variability in the data based on the genes. Unadjusted Cox regression analyses were conducted to determine the effect of each gene on the risk of mortality and recurrence. The results of Cox analyses were summarized with a hazard ratio (HR), 95% confidence interval (CI), and *P*-value. *P*-values less than 5% were considered statistically significant. All statistical analyses were performed using STATA 15.1 (StataCorp LLC, College Station, TX, USA).

## Results

We collected formalin-fixed paraffin-embedded tumors from 25 patients treated at our institution from 2012 to 2019. All patients had a pathologically confirmed diagnosis of stages II–IV TNBC and were treated at our institution and met the eligibility criteria. All tumor biopsies were performed before the treatment. Patients' demographic and clinical tumor characteristics are shown in Table 1. The majority of patients were postmenopausal at diagnosis, with an average age of 55.2 years. Twenty-four patients were HL (n = 24), and one patient was NHW. The majority of patients had stage 4 disease (84%) at the time of diagnosis. All selected patients received at least one line of chemotherapy, and seven (28%) received >2 lines of chemotherapy. Thirteen patients had a tumor proportion score (TPS) of more than 1%, and six patients received combined immunotherapy with chemotherapy (atezolizumab 840 mg D1, 15 Q21 days and nab-paclitaxel 100 mg/m$^2$ day 1, 8, and 15 Q 21 days cycle) at some point during their treatment. The most commonly mutated genes among HL women with TNBC were *TP53*, *NOTCH* genes, *AKT*, *MAP3K1*, *EGFR*, *PIK3CA*, and *PTEN* (Table 2).

**Table 1. Summary of demographic and clinicopathological characteristics of patients.**

| Characteristics | N (%) |
|---|---|
| Average age at diagnosis average | 56 |
| **Ethnicity/race** | |
| Hispanic Latino | 24 (94) |
| African American | 0 |
| Non-Hispanic white | 1 (6) |
| Other | 0 |
| **Histology** | |
| Invasive ductal carcinoma | 25 (100) |
| Mixed/other | 0 |
| **Clinical stage** | |
| IIIc/IV | 4 (16) |
| IV | 21(84) |
| **Chemotherapy lines** | |
| 2 | 13 (52) |
| 3+ | 7 (28) |

**Table 2. Proportions of the 10 most commonly mutated genes in Hispanic Latino women with metastatic triple-negative breast cancer at TTUHSC El Paso compared with other databases.**

| | TTUHSC | TCGA | COSMIC | METABRIC | (Chinese study) | (Thai study) |
|---|---|---|---|---|---|---|
| **Mutated** | **(n = 25)** | **(n = 93)** | **(n = 407)** | **(n = 159)** | **(n = 465)** | **(n = 116)** |
| **Gene** | **% (*P*-value)** | **% (*P*-value)** | **% (*P*-value)** | **% (*P*-value)** | **% (*P*-value)** | **% (*P*-value)** |
| *TP53* | **100** | 82.8 (**0.025**) | 51 (**<0.0001**) | 81.8 (**0.02**) | 74 (**0.003**) | 75.9 (**0.006**) |
| *NOTCH* | **44** | - - - | 2 (**<0.0001**) | 6.3 (**<0.0001**) | - - - | - - - |
| *AKT* | **28** | - - - | 1 (**<0.0001**) | 2.5 (**<0.0001**) | - - - | - - - |
| *MAP3K1* | **28** | - - - | n/r | 2.5 (**<0.0001**) | - - - | - - - |
| *PIK3CA* | 20 | 8.6 (0.106) | 10 (0.114) | 13.8 (0.415) | 18 (0.8) | 23.3 (0.721) |
| *EGFR* | 20 | - - - | 2 (**<0.0001**) | 1.9 (**<0.0001**) | - - - | - - - |
| *PTEN* | 16 | 1.08 (0.001) | 4 (0.006) | 6.3 (0.089) | 6 (0.048) | 11.2 (0.504) |

## Data comparison with other breast cancer databases

To demonstrate the diversity of tumor gene expression profiles, we further compared our data with more extensive international cancer databases such as METABRIC, COSMIC, and TCGA (with predominant NHW, and a small portion of AA women), and studies from China and Thailand (Asian population). Compared with the same tumor gene mutation frequencies from TCGA, COSMIC, METABRIC, and Chinese and Thai cohorts, HL patients with mTNBC had significantly higher mutation frequencies in *TP53*, *NOTCH* genes, *AKT*, *MAP3K1*, and *EGFR* (Table 2).

## Driver gene analysis

We examined the most frequently mutated genes to identify potential genes of interest. We first examined for genes mutated in multiple samples and found that 10 genes were mutated in at least 16% of the samples. *TP53* was the most frequently mutated gene, with variants found in 25 samples. The majority of the mutations were missense (single base substitution) at 76%, followed by frameshift mutations at 20%, and complete loss of *TP53* at 4%. The majority of *TP53* mutations (72%) were distributed in the DBD area in clusters within exons 5–8. One mutation was detected in exon 4. The rest of the mutations were distributed in exons 9–10 (Fig 1). Because *TP53* mutations were detected in all patients in our study, it was impossible to identify correlations with the outcomes. The previous data suggest that the prognostic effect of *TP53* was only limited to ER-positive breast cancer, particularly to worse outcomes in luminal B breast cancer [21]. In our study, however, we identified at least a trend for better outcomes for patients with *TP53* mutations in the tetrameric domain (exons 8–10). We also identified two most frequently mutated hotspots in *TP53*, at Y220C and I195T. Interestingly, the mutation in Y220C was described previously in patients with breast cancer and was associated with relatively favorable outcomes [22].

The second most common mutation observed in our study was in *NOTCH* genes, which encode transmembrane receptors that are highly conserved from invertebrates to mammals. In our study, we detected *NOTCH* gene mutations in 44% of analyzed tissue samples. Among all patients with mutated *NOTCH* genes, the most common alteration was in *NOTCH3* in 66.6% of patients, while the other types, *NOTCH1* and *NOTCH2*, were altered in 25% and 8.4%, respectively. Most alterations in *NOTCH1*, *NOTCH2*, and *NOTCH3* were located in the intracellular domain (58%), while one alteration was an amplification of *NOTCH2*, and the

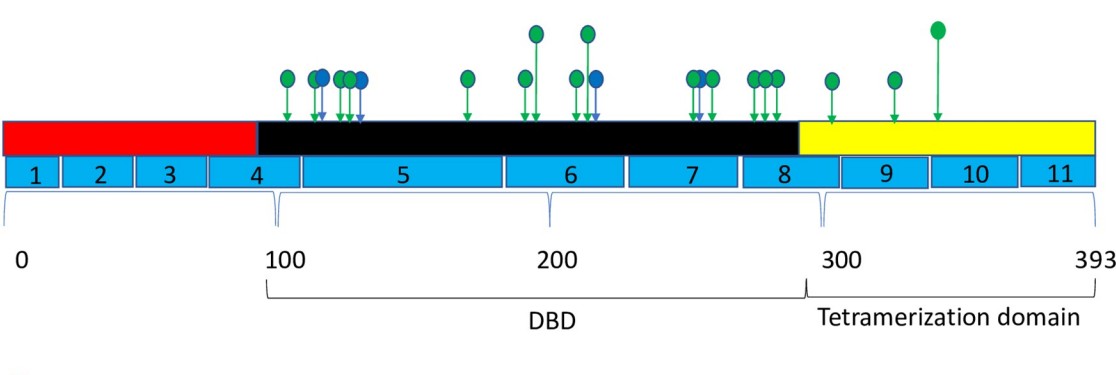

**Fig 1. Mutational spectrum of *TP53* in mTNBC.** The figure showed protein domains and the positions of specific mutations. A green dot indicated a missense mutation; and a blue dot indicated frameshift mutation.

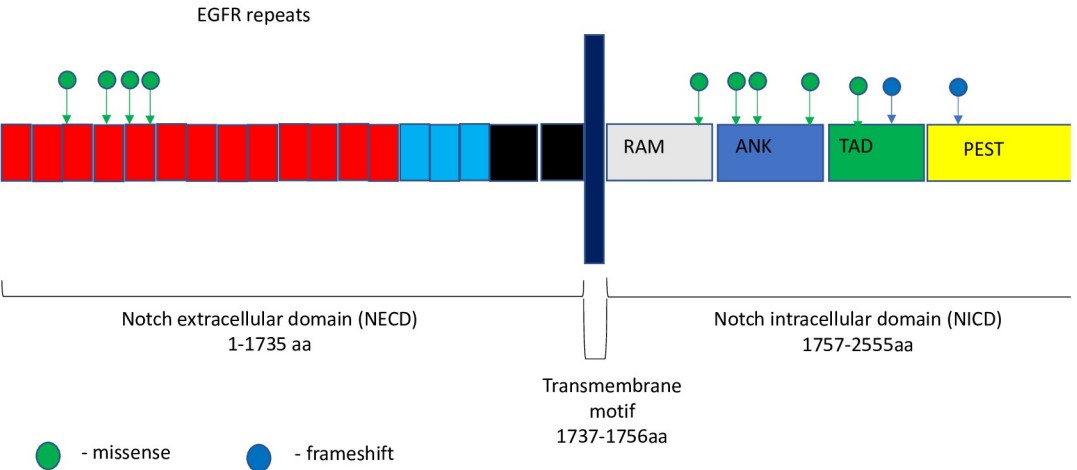

**Fig 2. Mutational spectrum of *Notch* mTNBC.** The figure showed protein domains and the positions of specific mutations. A green dot indicated a missense mutation; and a blue dot indicated frameshift mutation.

remaining mutations were located in the extracellular domain (Fig 2). All mutations in *NOTCH* genes were VUS except in three patients with rearrangements in exon 25, and in introns 24 and 18–26. At present, no data have described the role of any particular mutation in cancer progression. However, multiple studies showed that the most commonly mutated region in many types of cancer is the intracellular domain [23–25]. Interestingly, in the same studies, the authors demonstrated that the most common mutation in the intracellular domain was an activating mutation. However, the majority of mutations in the extracellular domain in *NOTCH* have been associated with a wide range of congenital disorders, such as bicuspid aortic valve, Alagille syndrome (a multisystemic disorder with cardiac, liver, ocular, and skeletal abnormalities), and cerebral arteriopathy [26].

The next most frequently mutated genes in the HL population were *MAP3K1* (28%), *AKT1/AKT2* (28%), *EGFR*, and *PIK3CA* (20%). Interestingly the frequency of mutations in *AKT*, *MAP3K*, and *EGFR* was significantly higher in the HL population compared with the patient populations reported in other databases. However, the mutations in *PIK3CA* and *PTEN* were in the same range (Table 2).

## Cluster analysis of specific variables

In our study, we demonstrated that mutations in *TP53* were present in all analyzed tissue samples—that is, 25 of the 25 samples (100%) containing a mutation. We did not identify any association between *TP53* and clinical outcomes or resistance to chemotherapy. The next most common mutation was in the *NOTCH* pathway, in 12 out of 25 samples. To understand the possible role of the most commonly mutated genes, such as *NOTCH* genes, *AKT*, *MAP3K1*, *EGFR*, *PIK3CA*, and *PTEN* and their association with outcomes and chemotherapy resistance, we performed a cluster analysis of specific variables (Table 3). The aggregation plot and Rand criteria indicated that three clusters retained maximum variability in the data based on genes. Our data revealed three clusters of genes: Cluster 1 included only *NOTCH* genes, Cluster 2 included three genes (*PTEN*, *AKT*, and *NF1*), and Cluster 3 included three genes (*EGFR*, *PIK3CA*, and *MAP3K1*). The most representative gene for Cluster 2 was identified as *PTEN*, and *MAP3K1* in Cluster 3 (Table 3).

**Table 3. Identification of cluster-specific variables.**

| Cluster | Cluster size | Mutated genes | Factor loading | Unique variances |
|---------|-------------|---------------|----------------|------------------|
| 1 | 1 | NOTCH | 1.00 | 1.00 |
| 2 | 2 | PTEN | 0.097 | 0.991 |
|   |   | AKT | −0.425 | 0.819 |
| 3 | 3 | EGFR | 0.845 | 0.285 |
|   |   | PIK3CA | 0.866 | 0.248 |
|   |   | MAP3K1 | 0.059 | 0.996 |

Factor loading: Presents the weight associated with each gene mutation within a cluster and is used for determining cluster score. Unique variance: A low value of unique variance associated with a gene mutation indicates a better predictive performance of that gene within a cluster. Cluster size: Provides the number of gene mutations within a cluster.

## Associations between genes and cluster of genes with PFS and OS

Patients with an increased number of gene mutations among 10 considered genes were associated with an increased risk of death (HR = 1.17, $P = 0.017$) and PFS (HR = 1.08, $P = 0.09$). Among identified clusters of gene mutations, Cluster 1 (*NOTCH*) showed significantly worse PFS among patients (HR 8.23, $P = 0.002$). Although not statistically significant, the presence of *NOTCH* gene mutations also increased the risk of death (HR = 3.89, $P = 0.11$). Cluster 3 (*MAP3K1*, *PIK3CA*, *EGFR*) was associated with an increased risk of mortality (HR = 3.80, $P = 0.007$). Our data demonstrated that the presence of either Cluster 1 (*NOTCH*) or Cluster 3 (*MAP3K1*, *PIK3CA*, *EGFR*) was strongly associated with OS ($P = 0.027$) as well as PFS ($P = 0.044$). Furthermore, patients with more mutated genes (*NOTCH* genes, *EGFR*, *PIK3CA*, *MAP3K1*) had a significantly higher risk of mortality (HR = 5.38, $P = 0.004$) as well as PFS (HR = 1.70, $P = 0.039$). Among individual genes, *NOTCH* genes (HR = 3.89, $P = 0.11$), *PIK3CA* (HR = 10.52, $P = 0.003$), *MAP3K1* (HR = 2.77, $P = 0.19$), and *EGFR* (HR = 3.61, $P = 0.10$) tended to be associated with an increased risk of mortality (Table 4).

**Table 4. Univariate Cox regression of overall survival (OS) and progression-free survival (PFS).**

| | OS | | PFS | |
|---|---|---|---|---|
| **Variable** | **HR (95% CI)** | **P-value** | **HR (95% CI)** | **P-value** |
| Age at diagnosis | 0.94 (0.89, 1.00) | 0.06 | 0.96 (0.92, 1.01) | 0.144 |
| Number of all genes | 1.17(1.03, 1.34) | 0.017 | 1.08 (0.98, 1.18) | 0.092 |
| Presence of any genes (*NOTCH*, *EGFR*, *PIK3CA*, *MAP3K1*) | 50% vs. 100% | 0.027 | 3.91 (1.04, 14.75) | 0.044 |
| Number of genes(*NOTCH*, *EGFR*, *PIK3CA*, *MAP3K1*) | 5.38(1.72, 16.83) | 0.004 | 1.70 (1.03, 2.82) | 0.039 |
| Factor 1 (*NOTCH*) | 3.89 (0.75, 20.23) | 0.106 | **8.23 (2.11, 32.02)** | **0.002** |
| Factor 2 (*PTEN*, *AKT2*, *NF1*) | 0.59 (0.04, 7.76) | 0.688 | **1.11 (0.16, 7.87)** | **0.0914** |
| Factor 3 (*EGFR*, *PIK3CA*, *MAP3K1*) | 3.80 (1.44, 10.05) | 0.007 | 1.35 (0.61, 3.02) | 0.458 |
| *PIK3CA* | **10.52 (2.23, 49.51)** | **0.003** | **2.00 (0.61, 6.59)** | **0.254** |
| *MAP3K1* | 2.77 (0.60, 12.74) | 0.19 | 1.18 (0.32, 4.44) | 0.797 |
| *EGFR* | 3.61 (0.79, 16.3) | 0.096 | 1.11 (0.31, 4.06) | 0.869 |
| *AKT* | 1.03 (0.19, 5.34) | 0.971 | 1.76 (0.56, 5.58) | 0.336 |
| *PTEN* | 0.79 (0.09, 6.6) | 0.829 | 1.92 (0.5, 7.31) | 0.338 |

## Discussion

In this study, we analyzed the individual tumor gene mutation profiles and their associations with clinical outcomes (PFS, OS) of HL women with TNBC. We identified a statistically significant frequency of mutations in some oncogenes. Notably, the most frequent mutation was in the *TP53* gene (100%). Besides *TP53* alterations, HL women in this study had tumors with significantly higher mutation rates in *NOTCH*, *AKT*, *EGFR*, and *MAP3K* compared with historical data from NHW, AA, and Asian women (Table 2). Our findings suggest that breast cancer driver mutations could be exceptionally different among different racial or ethnic groups, which could potentially explain some of the differences seen in the clinical outcomes between HL and other groups including NHW.

TNBC represents a heterogeneous subtype of breast cancer with adverse clinical outcomes and inconsistent responses to current therapy, particularly in advanced-stage disease [3, 4, 8]. Data from previous studies revealed that the spectrum of mutation profiles is diverse between each patient and also differs amongst racial and ethnic groups [10, 17, 18, 27, 28]. Although there are multiple explanations for the diverse genomic landscape of TNBC patients, ethnicity could have a significant role in this discrepancy. In this study, we identified mutations in *TP53* in all analyzed tissue samples.

*TP53* is widely considered to be a guardian of the genome because of its critical function in maintaining genome integrity, regulating the cell cycle, and initiating apoptosis. Multiple studies reported the frequency of *TP53* mutation in human breast cancer ranged from 50% to 82% [17–21].

This study represents a step forward in the field because the mutational profile of breast cancer in HL women has not been extensively analyzed to date, and limited data exist to enable a comparison with our results. Nevertheless, we identified two smaller studies (n = 19) from Northeast Mexico and another study from the National Cancer Institute of Mexico in Mexico City (n = 12) that analyzed data from a similar population of patients [29]. Importantly, the Northeast Mexico study reported the same frequency of *TP53* mutations (at 100%), supporting our findings [29]. The second study from the National Cancer Institute of Mexico showed the frequency of *TP53* mutations to be only 54% [30]. This difference might be explained by the inclusion of only patients with early-stage TNBC in the study from the National Cancer Institute of Mexico, unlike our study and the Northeast Mexico study. *TP53* is one of the most commonly mutated genes in multiple types of human cancer [21]. However, the frequency of *TP53* mutations significantly varies among different cancer types. For instance, TCGA database reported the highest frequency of *TP53* mutations in uterine cancer (90%), followed by 83% of mutations in NSCLC, ovarian, and esophageal cancer, 80% in colorectal cancer, and 72% in HNSCC.

In contrast, mutations are infrequent in thyroid cancer, occurring in only 2% of cases, followed by renal cell carcinoma and germ-cell tumors in approximately 1%, and 0.6%, respectively. While numerous published data demonstrated poor clinical outcomes for patients with mutated *TP53*, the exact role of *TP53* mutations in oncogenesis remains controversial [21]. Multiple theories describe the probable role of mutated p53 in oncogenesis. For instance, mutated p53 protein might serve as a negative inhibitor compared with wild-type p53 and therefore allow the proliferation of tumor cells [21]. Another theory suggests that mutated p53 gains a novel function, a "tumor transforming function," which gives tumor cells an advantage in uncontrolled proliferation [31]. This is based on the fact that the most common type of *TP53* alternation is a so-called missense mutation (62%). In the results, one amino acid (from the native protein) was replaced by a different amino acid (mutated protein). This can lead to the formation of abnormal p53 protein that can be functional and stably expressed in the tumor cells and might have a role in oncogenesis.

Over the past few decades, multiple compounds have been tested in clinical trials that target *TP53* by reactivating the mutated p53 protein and converting it to a conformation with wild-type properties, but at present, this approach remains experimental and no approved treatment option is available to address *TP53* mutation or loss [32]. Some promising compounds, especially AZD1775, APR-246, and COTI-2, have been found to exhibit anticancer activity in preclinical models of breast cancer [33–35].

In summary, *TP53* was found as the predominant mutation in HL women with mTNBC in this study, with missense mutations occurring in the DNA-binding domain. On the basis of our data and the currently available literature, including the Northeast Mexico study, we propose to further evaluate mutations in *TP53* as a likely driver mutation in HL women with TNBC. Additionally, because *TP53* mutations were detected in all patients in our study, and because of the small sample size, which was a limitation in our study, it was not feasible to identify correlations with the outcomes and chemotherapy resistance. However, *TP53* alterations are likely to have an important role in oncogenesis, and, along with the other mutations, might contribute to aggressive tumor behavior.

The second most common genetic alteration we noted in our HL patient population was mutations in the NOTCH pathway (44%). The NOTCH pathway regulates cell-fate decisions during embryogenesis [35]. NOTCH protein serves as a receptor for membrane-bound ligands Jagged 1, Jagged 2, and Delta 1. NOTCH receptors act in response to the ligands expressed by adjacent cells to regulate cell-fate specification, differentiation, proliferation, and survival [36, 37]. Multiple studies in the past have demonstrated upregulated expression of NOTCH receptors and their ligands in various human malignancies, such as colon, head and neck, lung, and breast cancer [23, 38, 39]. It was demonstrated in vitro that mutated *NOTCH* promotes the epithelial–mesenchymal transition (EMT) of MCF-10 cells and also protects transformed cells from p53-mediated apoptosis [40].

Furthermore, activated NOTCH pathway has an essential role in breast cancer cell migration and invasion [41]. A possible mechanism is NOTCH-mediated EMT. EMT occurs during tumor progression when cells from a primary epithelial tumor change phenotype, becoming mesenchymal, and disseminate as single metastatic cells to invade other organs. EMT may also be involved in the dedifferentiation program that leads to malignant carcinoma. Activation of endogenous NOTCH receptors in human endothelial cells was associated with EMT in endothelial cells, and upregulation of NOTCH in the MCF7 cell line promoted migratory transformation.

A meta-analysis of 3867 breast cancer patients demonstrated significantly worse OS and PFS in patients with upregulated *NOTCH1*. Among those patients, the most common subtype of breast cancer was the basal subtype [42]. In our study, we demonstrated significantly worse PFS among women with a mutated NOTCH pathway (HR 8.23; *P* = 0.002; Table 4). Importantly, mutations in NOTCH pathways were not described in HL women with TNBC in the current literature, and it was also quite a rare breast cancer mutation in the METABRIC, COSMIC, and TCGA databases (Table 2). Additional data about the role of a mutated NOTCH pathway in TNBC was reported in European and Chinese studies. For instance, Wang et al. described the prevalence of NOTCH4 pathway activation among Chinese women with TNBC. The authors demonstrated that the NOTCH4 pathway was upregulated in 55.6% of Chinese women with TNBC and was associated with a higher rate of recurrence [43]. Another study from Italy demonstrated that a higher level of *NOTCH1* expression is characteristic of a subclass of TNBC with poor outcomes. Patients with tumors expressing high levels of *NOTCH1* had worse OS compared with patients with low levels of expression (5-year survival rate was 49% versus 64%) [39]. Our data and data from other studies demonstrated that the alteration

of the NOTCH signaling pathway in TNBC is a critical event in tumorigenesis, and it is another possible driver mutation for this type of tumor [39, 40].

At present, there is limited data describing the role of any particular *NOTCH* mutation in cancer progression. Some data suggest that the majority of mutations in *NOTCH* are located in the intracellular domain. In one study, the majority of the intracellular mutations were activating mutations [44]. However, the majority of mutations in the extracellular domain of *NOTCH* have been associated with a wide range of congenital disorders [26].

Selective targeting of the mutated NOTCH pathway is a very attractive and promising treatment modality for such patients. NOTCH inhibitors and gamma-secretase inhibitors (GSIs) may be potential therapeutic approaches in the case of *NOTCH*-activating mutations [24, 39]. In theory, GSIs prevent proteolytic cleavage by inhibiting GSI activity and inhibiting the interaction of Jagged1 and NOTCH, thereby preventing endothelial activation [39]. NOTCH crosstalk between tumor cells, stromal cells, and endothelial cells regulates the interaction of NOTCH ligands on tumor cells with receptors on endothelial cells. Promising data were obtained in phase I/II clinical trials [45–48]. However, because of the high level of severe gastrointestinal toxicity from GSI treatment, multiple clinical trials have been postponed or terminated.

Another common mutation detected in our study was in the MAP3K1 pathway. MAP3K1 is a serine/threonine-protein kinase that acts as an essential upstream activator of mitogen-activated protein kinase (MAPK) signaling in response to stress. It was previously reported that an inactivating mutation in *MAP3K1*, together with one of its downstream substrates encoded by *MAP2K4*, was more prevalent in the luminal A subtype of breast cancer [49]. Moreover, it was previously demonstrated that MAP3K1 plays a role in cell migration and survival [50]. Interestingly, mutations in *MAP3K1*/*MAP2K4* are more prevalent in breast, prostate, and stomach cancer, and less common in other types of cancers [51]. There are no targeted therapies available to address genomic alterations in *MAP3K1* to date, but this area could be the basis for future research.

In our study, the frequency of *PIK3CA* and *PTEN* mutations was 20%, which is similar to other published breast cancer studies. The frequency of *AKT1*/*AKT2* mutations was 28%, which is significantly higher compared with other databases. Interestingly, some research suggests that *PIK3CA* and *AKT1* are mutually exclusive, but both can co-exist with *PTEN* mutations [52]. Mutations in *PTEN* were reported in multiple human malignancies in the past. Inactivation of *PTEN* leads to uncontrolled activation of the PIK3 pathway and cell proliferation. The most common mechanisms of *PTEN* inactivation are somatic mutations and monoallelic or biallelic deletion of the *PTEN* gene. However, some other mechanisms have been suggested, such as epigenetic silencing through promoter methylation, accelerated protein degradation, and post-translational modification [53, 54]. Interestingly, loss of *PTEN* heterozygosity was reported in 40%–50% of breast tumors, but functional inactivation of *PTEN* was reported in only 5%–10% of BC cases. The most common reported mechanism of *PTEN* inactivation is frameshift mutation [54]. As for other solid tumors, epigenetic mechanisms of *PTEN* modulation have also been reported for breast cancer [55].

Importantly, our study demonstrated significantly worse OS among patients with *PIK3CA* mutations (Table 4). However, we did not identify any major *PIK3CA* hotspot mutations among our patient population. *PIK3CA*/*AKT*/*PTEN* pathway mutations are usually enriched in hormonal receptor-positive tumors at 29%–45%, with slightly lower frequency in TNBC (TCGA).

The exact mechanism of the interaction between the PIK3K/APT/PTEN and NOTCH pathways is not well understood. Some recent data suggest that one mechanism may be connected with the downregulation of *PTEN* by activated NOTCH [56]. In some malignancies such as acute T-cell lymphoblastic leukemia, activation of the PI3K/AKT pathway downstream of NOTCH1 signaling promotes cell proliferation at multiple levels and has an important role

in T-cell transformation [56, 57]. Analysis of the transcriptional responses of GSI-sensitive *PTEN* wild-type glioblastoma cells to NOTCH inhibition showed significant upregulation of *PTEN* expression. The authors demonstrated one possible mechanism of *PTEN* downregulation in vitro, which was mediated by HES1-a transcriptional repressor directly controlled by *NOTCH* [57].

Mutations in *PIK3CA/AKT/PTEN* were evaluated as a potential target for appropriate inhibitors. For instance, in a randomized, placebo-controlled, phase II clinical trial, LOTUS, the combination of weekly paclitaxel with ipatasertib (AKT inhibitor) significantly improved PFS from 4.9 to 6.2 months in patients with mTNBC [58]. In a PAKT randomized- placebo-controlled clinical trial, another AKT inhibitor, capivasertib, was tested in combination with paclitaxel as first-line therapy in patients with mTNBC and demonstrated significantly better PFS and OS for patients with an altered PIK3CA/AKT/PTEN pathway [59].

Recently, the FDA granted approved for alpelisib, a new PI3K inhibitor, in combination with fulvestrant for patients with metastatic HR+/HER2− breast cancer based on the positive results of the SOLAR-1 trial. The study demonstrated significant activity of alpelisib in PIK3CA-mutant HR+/HER2− breast cancer compared with the placebo. The combination of fulvestrant with alpelisib improved PFS compared with fulvestrant with placebo (11 vs. 5.7 months, HR = 0.65) and ORR (26.6 vs. 12.8%) [60]. The BELLE-2 trial of endocrine-resistant HR+ breast cancer evaluated the combination of the pan-PI3K inhibitor buparlisib with fulvestrant. It demonstrated significantly increased PFS (7.0 vs. 3.2 months) and ORR (18% vs. 4%) with fulvestrant compared with placebo in patients with *PIK3CA* mutations [61]. Unfortunately, an attempt to adopt the same principle for the treatment of mTNBC was quite discouraging [62]. Although preclinical data suggest the efficacy of all PI3K/AKT/PTEN inhibitors alone or in combination with chemotherapy, current clinical evidence indicates that only AKT targeting in TNBC has the most efficiency in pathway-aberrant tumors. Of course, it will be essential to evaluate AKT inhibitors in future phase III trials as well as the combination of PI3K/AKT/PTEN inhibitors with immunotherapy.

In summary, in this study, we identified prevalent TNBC tumor mutations in HL patients, which could contribute useful information to the genomic landscape of breast cancer and provide more evidence to support the role of *TP53*, *NOTCH*, *MAP3K*, *AKT*, and *PIK3CA* in breast carcinogenesis.

This study had three significant limitations. First, we only enrolled patients from the area of El Paso in Texas, USA, Las Cruz in New Mexico, USA, and Juarez, Mexico. Thus, the data may not represent the entire HL population with mTNBC. Second, the small sample size and short follow-up period might compromise the observed, clinically meaningful association between genomic alterations and clinical outcomes and thus an investigation with a larger sample size is warranted in the future. Third, genomic sequencing was performed only on tumor DNA extracted from FFPE samples. It has been recognized that the quality of DNA from FFPE is lower than that of fresh samples and potentially causes variant call discrepancies. In this study, we focused on the list of cancer-associated genes and applied a variant call only when genomic regions had sufficient sequencing depth. This approach has been shown to minimize erroneous variant calls, improve precision, and enable acceptable correlations with matched normal-tumor pair sequencing. Nevertheless, comparison of our data with previously published studies could be limited by differences in study designs and data analysis methods.

## Conclusions

This study represents one of the first studies of HL women with mTNBC focusing on distinctive genomic alterations. Significantly higher mutation frequencies were noted in several

cancer-associated genes, notably *TP53*, *NOTCH*, *AKT*, *EGFR*, *MAP3K*, and *PIK3CA*. Importantly the presence of mutations in *NOTCH* and *PIK3CA*, individually or in combination, was associated with worse outcomes (OS as well as PFS). These results support the genomic heterogeneity between NHW, AA, Asian, and HL individuals, and, if confirmed in larger trials, could contribute to improvements in the diagnostic and therapeutic approaches for these patients.

## Acknowledgments

The authors would like to acknowledge Sean Connery, Luis Alvarado, Brenda Castillio, and Rosalinda Heydarian for their efforts in the article preparation. We also thank H. Nikki March, PhD, from Edanz Group (https://en-author-services.edanzgroup.com/) for editing a draft of this manuscript.

## Author Contributions

**Conceptualization:** Alexander Philipovskiy, Sumit Gaur.

**Data curation:** Alexander Philipovskiy.

**Formal analysis:** Roberto Gamez, Sumit Gaur.

**Investigation:** Alexander Philipovskiy, Renato J. Aguilera.

**Methodology:** Alexander Philipovskiy, Alok K. Dwivedi, Roberto Gamez, Renato J. Aguilera.

**Project administration:** Alexander Philipovskiy.

**Resources:** Renato J. Aguilera.

**Software:** Alok K. Dwivedi.

**Supervision:** Alexander Philipovskiy, Richard McCallum, Debabrata Mukherjee, Zeina Nahleh, Renato J. Aguilera, Sumit Gaur.

**Validation:** Alexander Philipovskiy, Alok K. Dwivedi.

**Visualization:** Alexander Philipovskiy.

**Writing – original draft:** Alexander Philipovskiy.

**Writing – review & editing:** Alexander Philipovskiy, Alok K. Dwivedi, Richard McCallum, Debabrata Mukherjee, Zeina Nahleh, Renato J. Aguilera, Sumit Gaur.

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
