## [Decision Letter · Decision Letter 0]

17 Jun 2020

PONE-D-20-15204

Association between tumor mutation profile and clinical outcomes among Hispanic Latina women with triple-negative breast cancer.

PLOS ONE

Dear Dr. Philipovskiy,

Thank you for submitting your manuscript to PLOS ONE. After careful consideration, we feel that it has merit but does not fully meet PLOS ONE’s publication criteria as it currently stands. Therefore, we invite you to submit a revised version of the manuscript that addresses the points raised during the review process.

Please respond to each and every comments (major and minor) to satisfy both the reviewers. Please consider increasing the sample size as the strength of the data (conclusion based on it) depends to the size of the cohort.

We look forward to receiving your revised manuscript.

Kind regards,

Nandini Dey, MS., Ph.D

Academic Editor

PLOS ONE

Journal Requirements:

2. In the ethics statement in the manuscript and in the online submission form, please provide additional information about the patient records used in your retrospective study. Specifically, please ensure that you have discussed whether all data/tissue samples  were fully anonymized before you accessed them and/or whether the IRB or ethics committee waived the requirement for informed consent. If patients provided informed written consent to have data from their medical records used in research, please include this information.

3. We noticed minor instances of text overlap with the following previous publication(s), which need to be addressed:

(1) http://atlasgeneticsoncology.org/Genes/NOTCH1ID30ch9q34

(2) https://emedicine.medscape.com/article/1372666-overview?cc=aHR0cDovL2VtZWRpY2luZS5tZWRzY2FwZS5jb20vYXJ0aWNsZS8xMzcyNjY2LW92ZXJ2aWV3&cookieCheck=1

(3) https://peerj.com/articles/6501/

The text that needs to be addressed involves the Discussion section.

In your revision please ensure you cite all your sources (including your own works), and quote or rephrase any duplicated text outside the methods section. Further consideration is dependent on these concerns being addressed.

4. Please provide the accession numbers or specific weblinks to the specific datasets obtained from public databases analyzed in this study.

5. We suggest you thoroughly copyedit your manuscript for language usage, spelling, and grammar. If you do not know anyone who can help you do this, you may wish to consider employing a professional scientific editing service.  

7. PLOS requires an ORCID iD for the corresponding author in Editorial Manager on papers submitted after December 6th, 2016. Please ensure that you have an ORCID iD and that it is validated in Editorial Manager. To do this, go to ‘Update my Information’ (in the upper left-hand corner of the main menu), and click on the Fetch/Validate link next to the ORCID field. This will take you to the ORCID site and allow you to create a new iD or authenticate a pre-existing iD in Editorial Manager. Please see the following video for instructions on linking an ORCID iD to your Editorial Manager account: https://www.youtube.com/watch?v=_xcclfuvtxQ

Reviewers' comments:

Reviewer's Responses to Questions

**Comments to the Author**

1. Is the manuscript technically sound, and do the data support the conclusions?

Reviewer #1: Yes

Reviewer #2: Yes

2. Has the statistical analysis been performed appropriately and rigorously? 

Reviewer #1: I Don't Know

Reviewer #2: I Don't Know

3. Have the authors made all data underlying the findings in their manuscript fully available?

Reviewer #1: Yes

Reviewer #2: Yes

4. Is the manuscript presented in an intelligible fashion and written in standard English?

Reviewer #1: Yes

Reviewer #2: Yes

5. Review Comments to the Author

Reviewer #1: This is a nice and important MS in the era of precision medicine especially with TNBC where no obvious biomarker(s) is available for targeted therapy. Hispanic women's with TNBC-specific genomic data is rare. In this aspect this is an important article. But it needs few revisions.

Critical comments:

1. Sample size is small. Is it possible to add more samples either from parent institute or from their collaborative institute?

2. It is always better if authors provide some germline alterations data along with somatic alterations?

3. All ECD mutations of NOTCH1 are in VUS or not? Please specify it.

4. Authors may provide data of INPP4B (one of the important phosphates in the PI3K pathway) and its frequency of alterations in TNBC is almost 30% (Nature 2012). It is good for the readers if authors will add some data.

5. Their data showed PTEN mutation rate is 16%. Mutation per se it is high compared to regular TNBC data ( Nature 2012 and Cancer Discovery 2013). But total alterations (mutation/silencing/loss) it is more than 30%. Please discuss this issue in the discussion section.

6. Table 3 needs some description.

7. Discussion is way to long. For example, authors may take out the paragraph of TP53 inhibitor part, same for NOTCH and EMT section as well as BELLE2 trial data.

7. Authors may provide the overlap of alterations of PI3K pathway and NOTCH genes. It will strengthen the MS.

8. Authors mentioned AKT mutations was 26%. Is it AKT1 or AKT2 or AKT3? Please specify.

Minor comments;

Typo error

Need some figure legend for Figure 1 and 2.

Reviewer #2: This is a very important and timely manuscript. The tumor mutation profiles and outcomes among Hispanic women with triple-negative breast cancer (TNBC) is under studied. Authors put together a very good manuscript using Hispanic-Latina (HL) TNBC patients data. I do have some comments and suggestions to make this manuscript comprehensive. These are as follows:

Abstract:

This sentence needs to be revised: “The aim of the study was to characterize individual patient gene expression profiles and to identify the relationship with clinical outcomes”. They have done mutational analysis.

Need some references in Introduction section:

e.g.

Importantly, multiple studies suggest that the prevalence of TNBC among HL women can be slightly higher compared to NHW, approaching 23.1%. Also, the onset of the disease occurs at age approximately 11 years younger than the average age reported for NHW and AA women.

Methods:

It is mentioned that patients were stage II-IV. Please confirm metastatic status of all samples.

Please clarify the rationale of selecting 24 HL out of 25 TNBC patients.

Describe the control samples used for this study.

Results:

Authors described Notch1-3. How about Notch 4? In the discussion section, authors talked about the role of Notch 4 from another paper.

Table: 1

Tell us about the effect of Immunotherapy treatments. Which immunotherapy was used and which combinations?

Table 3: Rationale for the three clusters.

Discussion:

We all know the importance of p53 in TNBC and other breast cancers. This manuscript talks extensively on Notch mutations. I would like to see the Notch expression compared to other HL patients in the cited paper (17 and 25).

Many GSI related clinical trials have been postponed or terminated. Please address those as well. One of the major issues of GSI is that it causes severe intestinal toxicity; more importantly, Notch is required for T cell functions as well.

Spelling check : Innovatibe, datat

Need references: “Nevertheless, we identified two smaller studies (n=19) from Northeast Mexico, and another study from the National Cancer Institute of Mexico in Mexico City (n=12) analyzing datat from a similar population of patients”.

6. PLOS authors have the option to publish the peer review history of their article (what does this mean?). If published, this will include your full peer review and any attached files.

Reviewer #1: No

Reviewer #2: No

---

## [Author Response · Author response to Decision Letter 0]

3 Aug 2020

Response to the academic editor 

Dear Dr. Nandini Dey 

Thank you very much for your review of our paper. We greatly appreciate the time you have spent on your careful review of our manuscript. We are grateful for your thoughtful comments and constructive suggestions, which have helped us to improve the quality of our manuscript. 

Please find our responses below (reviewers’ comments are in italics): 

Editor: When submitting your revision, we need you to address these additional requirements.

 1. Please ensure that your manuscript meets PLOS ONE's style requirements, including those for file naming. The PLOS ONE style templates can be found at: https://journals.plos.org/plosone/s/file?id=wjVg/PLOSOne_formatting_sample_main_body.pdf and https://journals.plos.org/plosone/s/file?id=ba62/PLOSOne_formatting_sample_title_authors_affiliations.pdf

Reply: We apologies for this error; it has been corrected.

Editor: 2. In the ethics statement in the manuscript and in the online submission form, please provide additional information about the patient records used in your retrospective study. Specifically, please ensure that you have discussed whether all data/tissue samples were fully anonymized before you accessed them and/or whether the IRB or ethics committee waived the requirement for informed consent. If patients provided informed written consent to have data from their medical records used in research, please include this information.

 Reply: We appreciate your recommendation and have added an appropriate statement (lines 103–104). 

Editor: 3. We noticed minor instances of text overlap with the following previous publication(s), which need to be addressed:

(1) http://atlasgeneticsoncology.org/Genes/NOTCH1ID30ch9q34

(2) https://emedicine.medscape.com/article/1372666-overview?cc=aHR0cDovL2VtZWRpY2luZS5tZWRzY2FwZS5jb20vYXJ0aWNsZS8xMzcyNjY2LW92ZXJ2aWV3&cookieCheck=1

Reply: We apologies for this error, but we are not quite sure which part of our paper is similar to the abovementioned links, is it possible to clarify and we will be more than happy to address. 

(3)https://peerj.com/articles/6501/

this link was appropriately cited (lines 523-525) 18.Niyomnaitham S, Parinyanitikul N, Roothumnong E, Jinda W, Samarnthai N, Atikankul T, et al. Tumor mutational profile of triple negative breast cancer patients in Thailand revealed distinctive genetic alteration in chromatin remodeling gene. PeerJ. 2019; 7: e6501. doi: 10.7717/peerj.6501.

Editor: 4. Please provide the accession numbers or specific weblinks to the specific datasets obtained from public databases analyzed in this study.

Reply: As suggested by the reviewer, we have added appropriate links (lines 134, 136-139, 141).

 Editor: 5. We suggest you thoroughly copyedit your manuscript for language usage, spelling, and grammar. If you do not know anyone who can help you do this, you may wish to consider employing a professional scientific editing service. 

Reply: We appreciate your recommendation, and an appropriate revision was requested from the Edanz Group (https://en-author-services.edanzgroup.com/) and performed by H. Nikki March, Ph.D. 

Editor: 6. We note that you have included the phrase “data not shown” in your manuscript. Unfortunately, this does not meet our data sharing requirements. PLOS does not permit references to inaccessible data. We require that authors provide all relevant data within the paper, Supporting Information files, or in an acceptable, public repository. Please add a citation to support this phrase or upload the data that corresponds with these findings to a stable repository (such as Figshare or Dryad) and provide and URLs, DOIs, or accession numbers that may be used to access these data. Or, if the data are not a core part of the research being presented in yo

 Reply: As suggested by the reviewer, we have addressed that error. (lines 171-175)

Editor: 7. PLOS requires an ORCID iD for the corresponding author in Editorial Manager on papers submitted after December 6th, 2016. Please ensure that you have an ORCID iD and that it is validated in Editorial Manager. To do this, go to ‘Update my Information’ (in the upper left-hand corner of the main menu), and click on the Fetch/Validate link next to the ORCID field. This will take you to the ORCID site and allow you to create a new iD or authenticate a pre-existing iD in Editorial Manager.

Reply: We appreciate your recommendation, the ORCID for the corresponding author is ORCID 0000-0002-7065-9578,

Response to reviewer #1 

Reviewer #1: This is a nice and important MS in the era of precision medicine especially with TNBC where no obvious biomarker(s) is available for targeted therapy. Hispanic women's with TNBC-specific genomic data is rare. In this aspect this is an important article. But it needs few revisions.

Reply: Thank you very much for your positive feedback on our paper. We greatly appreciate the time you have spent on your careful review of our manuscript. We are grateful for your thoughtful comments and constructive suggestions, which have helped us to improve the quality of our manuscript. Please find our responses below (reviewers’ comments are in italics): 

Critical comments:

 Sample size is small. Is it possible to add more samples either from parent institute or from their collaborative institute?

Reply: We appreciate your remark concerning the sample size in our study. It is very difficult to include additional patients in the current study because of its retrospective nature and the specific outcomes that we analyzed, such as progression-free and overall survival, as well as the specific population of patients. 

We are currently planning a prospective study and aim to recruit a larger group of patients in the future. 

 It is always better if authors provide some germline alterations data along with somatic alterations? 

Reply: Thank you for your suggestion. We routinely test all appropriate cases for germline mutations. However, not all patients in the study were eligible based on the current criteria recommended by the NCCN. Among tested patients, the only one had a BRCA1 germline mutation, while three others had variants of uncertain significance (VUS) in CDKN2A, NBN, and MSH6. Therefore, we did not include this information in our manuscript. However, if Reviewer #1 would recommend the inclusion of this information, then we would be more than happy to do so. 

All ECD mutations of NOTCH1 are in VUS or not? Please specify it.

Reply: Apologies for this error, this has been corrected (lines 197–198). All mutations in NOTCH1 were VUS except one patient with a rearrangement in exon 25. 

 Authors may provide data of INPP4B (one of the important phosphates in the PI3K pathway) and its frequency of alterations in TNBC is almost 30% (Nature 2012). It is good for the readers if authors will add some data. 

Reply: Interestingly, in contrast to the study published in Nature in 2012 that was mentioned in our report, only one patient had an INPP4B mutation, which might support our general theory about different driver mutations or mechanisms in cancer progression among HL women. 

Their data showed PTEN mutation rate is 16%. Mutation per se it is high compared to regular TNBC data ( Nature 2012 and Cancer Discovery 2013). But total alterations (mutation/silencing/loss) it is more than 30%. Please discuss this issue in the discussion section.

Reply: We appreciate your recommendation and have added a paragraph discussing PTEN to the revised manuscript. (lines 421–442). 

Table 3 needs some description. 

Reply: As suggested by the reviewer, we have added a description for Table 3 to the footnote as well as to the main text (lines 155–157 and 215), as follows: 

Factor loading: Presents the weight associated with each gene mutation within a cluster and is used for determining cluster score. Unique variance: A low value of unique variance associated with a gene mutation indicates a better predictive performance of that gene within a cluster. Cluster size: Provides the number of gene mutations within a cluster.

Discussion is way to long. For example, authors may take out the paragraph of TP53 inhibitor part, same for NOTCH and EMT section as well as BELLE2 trial data. 

Reply: We agree with your point and have removed the abovementioned paragraphs from the Discussion. 

Authors may provide the overlap of alterations of PI3K pathway and NOTCH genes. It will strengthen the MS.

Reply: We appreciate your recommendation and have added a paragraph discussing the possible interplay between the PIK3/ATK/PTEN and NOTCH pathways (lines 440–446). 

Authors mentioned AKT mutations was 26%. Is it AKT1 or AKT2 or AKT3? Please specify.

Reply: Thank you for requesting further clarification. In our study, we only detected mutations in AKT1 (four patients) and AKT2 (three patients) (lines 206, 419). 

Minor comments:

Typo error

Reply: Thank you for your comment. We have reviewed and addressed this. 

Need some figure legend for Figure 1 and 2.

Reply: As suggested by the reviewer, we have added legends to Figure 1 and 2. 

Response to reviewer # 2 

Thank you very much for your positive feedback on our paper. We greatly appreciate the time you have spent in your careful review of our manuscript. We are grateful for your thoughtful comments and constructive suggestions, which have greatly improved the quality of this paper. Please find our responses below (reviewers’ comments are in italics): 

Reviewer #2: This is a very important and timely manuscript. The tumor mutation profiles and outcomes among Hispanic women with triple-negative breast cancer (TNBC) is under studied. Authors put together a very good manuscript using Hispanic-Latina (HL) TNBC patients data. I do have some comments and suggestions to make this manuscript comprehensive. These are as follows:

Abstract:

This sentence needs to be revised: “The aim of the study was to characterize individual patient gene expression profiles and to identify the relationship with clinical outcomes”. They have done mutational analysis.

Reply: As suggested by the reviewer, we have revised this sentence (line 34). 

Need some references in Introduction section: Importantly, multiple studies suggest that the prevalence of TNBC among HL women can be slightly higher compared to NHW, approaching 23.1%. Also, the onset of the disease occurs at age approximately 11 years younger than the average age reported for NHW and AA women.

Reply: As suggested by the reviewer, we have added additional references [12-14], and [1, 11] to support our statement. 

Methods: It is mentioned that patients were stage II-IV. Please confirm metastatic status of all samples. Please clarify the rationale of selecting 24 HL out of 25 TNBC patients. Describe the control samples used for this study.

Reply: In this study, we included patients with newly diagnosed stage IV (metastatic) TNBC, as well patients who were initially diagnosed with stages II–III and who progressed to stage IV during the 12-month observation period after the definitive treatment (surgery, chemotherapy, and radiation therapy). We did not have any control samples in the study because the original study was designed as a retrospective study without any interventions. We only compared HL patients with TNBC from national and international databases. We analyzed the tumor mutation profile of all 25 patients and highlighted that one patient was not HL. 

Results:

Authors described Notch1-3. How about Notch 4? In the discussion section, authors talked about the role of Notch 4 from another paper. 

Reply: We did not detect any NOTCH4 mutations in our patient population. 

Table: 1

Tell us about the effect of Immunotherapy treatments. Which immunotherapy was used and which combinations? 

Reply: Since we did not observe any statistically significant difference in PFS or OS following immunotherapy, and also because the sample size was too small for further analyses, we decided to remove this information from Table 1. For eligible patients, we used a standard of care treatment protocol (IMpassion 130 randomized phase III clinical trial) approved by the FDA on 03/08/2019: combined chemotherapy nab-paclitaxel 100 mg/m2 D1;8;15 Q 21 days cycle with anti-PDL-1 agent atezolizumab 840 mg D1;15 Q21 days(line 174). 

Table 3: Rationale for the three clusters.

Reply: We performed a variable cluster analysis for categorical variables and determined the number of clusters using an aggregation plot and mean adjusted Rand criteria, which indicated that three clusters retained maximum variability in the data based on the genes. This was a very useful analysis in this study, as several mutated genes might interact with each other. By cluster analysis, we were able to evaluate the joint effect of two or more gene mutations on overall survival and progression-free survival. We have added this information to the revised manuscript(lines 157–160).

Discussion: We all know the importance of p53 in TNBC and other breast cancers. This manuscript talks extensively on Notch mutations. I would like to see the Notch expression compared to other HL patients in the cited paper (17 and 25).

Reply: Thank you for this important point. However, in our study, we did not perform gene expression analysis. Therefore, we were unable to compare our data with published studies. Furthermore, we did not find any data for Notch expression in Hispanic or Asian patients in Jiang Z et al. (Genomic and Transcriptomic Landscape of Triple-Negative Breast Cancers: Subtypes and Treatment Strategies.) (ref 17). In addition, in Wang K et al., while the authors described activating mutations in NOTCH1, NOTCH2, and NOTCH3 and extensively discussed in vitro experimental data as well performing as some comparisons with TCGA database, they did not analyze and/or discuss any data for HL. 

Many GSI related clinical trials have been postponed or terminated. Please address those as well. One of the major issues of GSI is that it causes severe intestinal toxicity; more importantly, Notch is required for T cell functions as well.

Reply: We agree with your remark, and have added a relevant sentence dealing with this issue to the revised manuscript (line 409). We also decided to remove the discussion about Notch inhibition in clinical trials. Unfortunately, the results of multiple phase I trials showed some disappointments of Notch pathway inhibition. These issues were not only because of GI toxicity but also because of low response rates. 

Furthermore, among multiple previous studies, only a few showed some efficacy of Notch inhibition. In 2019, the FDA granted orphan drug designation to AL101 for the treatment of patients with adenoid cystic carcinoma with activating mutations in Notch. A phase II study (ACCURACY) is still recruiting patients. 

Spelling check:

Innovatibe,datat

Reply: The suggested corrections have been made and a draft of the revised manuscript has been edited by a native English editor. 

Need references: “Nevertheless, we identified two smaller studies (n=19) from Northeast Mexico, and another study from the National Cancer Institute of Mexico in Mexico City (n=12) analyzing data from a similar population of patients”.

Reply: As suggested by the reviewer, we have added these references to the revised manuscript ([29] and [30]).

---

## [Editor Report · Decision Letter 1]

6 Aug 2020

PONE-D-20-15204R1

Association between tumor mutation profile and clinical outcomes among Hispanic Latina women with triple-negative breast cancer.

PLOS ONE

Dear Dr. Philipovskiy,

Thank you for submitting your manuscript to PLOS ONE. After careful consideration, we feel that it has merit but does not fully meet PLOS ONE’s publication criteria as it currently stands. Therefore, we invite you to submit a revised version of the manuscript that addresses the points raised during the review process.

Please state in the discussion of the MS acknowledging that the low sample size is one of the limitations of the study, and thus an investigation with a larger sample size is warranted in the future.

We look forward to receiving your revised manuscript.

Kind regards,

Nandini Dey, MS., Ph.D

Academic Editor

PLOS ONE

---

## [Author Response · Author response to Decision Letter 1]

10 Aug 2020

Response to the academic editor: 

Dear Dr. Philipovskiy,

Thank you for submitting your manuscript to PLOS ONE. After careful consideration, we feel that it has merit but does not fully meet PLOS ONE’s publication criteria as it currently stands. Therefore, we invite you to submit a revised version of the manuscript that addresses the points raised during the review process.

Please state in the discussion of the MS acknowledging that the low sample size is one of the limitations of the study, and thus an investigation with a larger sample size is warranted in the future.

Reply: We appreciate your remark concerning the sample size in our study. As you recommend, we have added an appropriate statement (lines 440-442). 

Sincerely, 

Alexander Philipovskiy

---

## [Editor Report · Decision Letter 2]

13 Aug 2020

Association between tumor mutation profile and clinical outcomes among Hispanic Latina women with triple-negative breast cancer.

PONE-D-20-15204R2

Dear Dr. Philipovskiy,

We’re pleased to inform you that your manuscript has been judged scientifically suitable for publication and will be formally accepted for publication once it meets all outstanding technical requirements.

Kind regards,

Nandini Dey, MS., Ph.D

Academic Editor

PLOS ONE
---

## [Editor Report · Acceptance letter]

24 Aug 2020

PONE-D-20-15204R2 

Association between tumor mutation profile and clinical outcomes among Hispanic Latina women with triple-negative breast cancer 

Dear Dr. Philipovskiy:

I'm pleased to inform you that your manuscript has been deemed suitable for publication in PLOS ONE. Congratulations! Your manuscript is now with our production department. 

Kind regards, 

on behalf of

Dr. Nandini Dey 

Academic Editor

PLOS ONE